# Aggressive Vertebral Hemangioma and Spinal Cord Compression: A Particular Direct Access Case of Low Back Pain to Be Managed—A Case Report

**DOI:** 10.3390/ijerph192013276

**Published:** 2022-10-14

**Authors:** Fabrizio Brindisino, Angelo Scrimitore, Denis Pennella, Francesco Bruno, Raffaello Pellegrino, Filippo Maselli, Francesco Lena, Giuseppe Giovannico

**Affiliations:** 1Department of Medicine and Health Science “Vincenzo Tiberio”, University of Molise, 86100 Campobasso, Italy; 2Antalgic Mini-Invasive and Rehab-Outpatients Unit, Department of Medicine and Science of Aging, University “G. d’Annunzio” Chieti-Pescara, 66100 Chieti, Italy; 3Department of Scientific Research, Campus Ludes, Off-Campus Semmelweis University, 6912 Lugano, Switzerland; 4Department of Human Neurosciences, University of Rome “Sapienza”, 00185 Rome, Italy; 5Sovrintendenza Sanitaria Regionale Puglia INAIL, 70121 Bari, Italy; 6Department of Neurology, IRCCS INM Neuromed, 86077 Pozzilli, Italy

**Keywords:** hemangioma, physical therapy modalities, therapeutic embolization, neoplasm

## Abstract

Hemangiomas are the most common benign tumours affecting the spine, with an incidence of 10–12% of the general population. Although most hemangiomas are asymptomatic, there are aggressive forms which can develop symptoms, leading patients to show signs of disability. This case report aims to highlight the importance of red flags screening, and to report the physiotherapist’s clinical reasoning that led him to refer his patient to other healthcare professionals. This case also illustrated the pre- and post-surgical treatment of a specific low back pain case in a patient affected by aggressive vertebral hemangioma and spinal cord compression. The patient is a 52-year-old man, who reported intense pain in his sacral region about three months prior, which worsened while in sitting position. The physiotherapist proceeded with a complete medical history investigation and clinical examination. After an impaired neurological examination, the patient was referred to another health professional, who diagnosed multiple vertebral hemangiomas in the patient’s lumbosacral tract. The therapeutic intervention included the patient’s post-surgical rehabilitation following a vascular embolization. This case report shows the importance of proper patient screening. Indeed, during patients’ assessment, it is paramount to recognize red flags and to investigate them appropriately. An early referral of patients with conditions that require the support and expertise of other professionals can lead to a timely diagnosis and avoid costly and unnecessary rehabilitation procedures. In this case, the interdisciplinary collaboration between physiotherapist and neurosurgeon was crucial in guiding the patient towards recovery.

## 1. Introduction

Angiomatous lesions [1] of the musculoskeletal system are common and can affect both bone and soft tissue. Hemangiomas (HAs) are the benign tumours that most frequently affect the spine [2] and are pathologically classified by the predominant type of vascular canal (capillary, cavernous, arteriovenous or venous) involving endothelial cells that grow in the medulla of the vertebral body [3].

Most HAs are discovered accidentally in asymptomatic patients. Men are affected twice as often as women and lesions are usually discovered between the fourth and fifth decade of life [2]; moreover, bone HAs are particularly common in the spine and less frequently affect long bones, such as tibia, femur and humerus. Notably, vertebral HAs account for 28% of total skeletal tumours [4], with the thoracic spine as the most common site [4,5,6]. 

Vertebral HAs have an incidence of 10–12% in the general population [7,8] and show a coarse structure, such as vertical trabecular pattern, with bony reinforcement (thickened vertebral trabeculae) adjacent to the vascular channels that caused bone resorption [8] as the main radiographic finding. Using computed tomography (CT), the thickened vertebral trabeculae are visualized in cross-section as small spotted areas of sclerosis, often referred to as having a “polka-dot appearance”; using magnetic resonance imaging (MRI) T2-weighted images usually show areas of high signal intensity, corresponding to vascular components [4,9,10,11,12,13,14].

Rarely, in 0.9–1.2% of cases, HAs are defined as aggressive due to spread of the spinal cord within the soft tissues and epidural region, causing radicular and/or spinal cord compression, expansion of the bone matrix, compression of large vessels due to angiogenesis, epidural haematoma or spinal instability, caused by vertebral compression fracture [7,10,15] which may cause pain, paraesthesia and other neurological signs [14].

It is generally accepted that surgery is warranted when vertebral HAs cause severe back pain and neurological diseases; however, the optimal treatment strategy is controversial [16,17]. Indeed, some surgeons opt for operations such as vertebroplasty, endovascular embolization (EVE) or percutaneous embolization, ethanol injection or radiotherapy; EVE is particularly effective in cases where the pathogenetic mechanism of compression is caused by the extension of the lesion into the epidural space—rather than by vertebral swelling—and has also been proposed as a definitive treatment for aggressive vertebral HAs without surgery [7].

Physiotherapists (PTs) are among the health professionals most frequently required in the management of pain and disabilities related to musculoskeletal disorders [18] and patients with low back pain (LBP) may also self-refer to a PT in direct access.

Therefore, PTs must be adequately trained in order to be able to recognize clinical findings and symptoms, which may indicate an underlying extra professional pathology, suggesting the need for further medical investigation [12].

This case report aims to highlight the importance of red flags screening, in particular in a direct access environment; moreover, this case report describes the clinical reasoning that led the PT to refer a patient with LBP which was caused by aggressive HAs, thus requiring expertise beyond that of the PT. The patient was referred to additional health professionals for a rapid, in-depth assessment and appropriate management of his condition. 

## 2. Case Description

This case report was written following the Care guidelines [10], and the authors received written informed clinical consent from the patient. A synopsis of this episode of care is provided in the Timeline (Figure 1).

The 52-year-old male patient was examined at the authors’ private practice complaining of intense sacral pain, especially while in a sitting position (Numeric Pain Rating Scale–NPRS 8/10 [19]).

The patient worked as a bus driver 8–9 h a day and he spent his spare time at home, resting on the sofa and rarely taking short walks. He was a smoker (12 cigarettes/day) and he had a BMI of 32 (grade 1 obesity).

When questioned about his symptoms, the patient stated that 3 months prior he had started to feel a nagging pain in his sacral area (3/10 NPRS). The onset was described as insidious and without any trauma. The patient self-managed his pain through the use of non-steroidal anti-inflammatory drugs (Ibuprofen twice a day) and lumbar supports at work. However, the patient’s pain worsened day by day and became constant, severe and disabling (7/10 NPRS), until he was no longer able to work. Moreover, the patient could not sleep due to his pain and, following the advice of one of his colleagues, he asked for a PT consultation.

The PT deeply investigated the patient’s medical history: significant features were a slight but unjustified weight loss during the past 6 weeks (9 kg) and a report of testicular seminoma 5 years earlier (his testicle was excised in 2018, but the patient stopped his medical follow-up due to the COVID-19 pandemic in 2020).

The PT carried out a symptom review and the patient reported that, for two weeks, whenever he coughed or sneezed, he leaked some drops of urine; furthermore, saddle dysesthesia was reported (Figure 2). No other significant features nor relevant family history were reported.

## 3. Clinical Examination

A comprehensive patient assessment was performed to establish whether it was appropriate to treat, treat and report or simply refer the patient [20].

The physical examination began with frontal, lateral and posterior inspections and no altered features were noted. Superficial palpation revealed no significant abnormality. There was no skin scaring, deformity or discoloration.

The PT requested active trunk movements in a sitting position. Right and left lateral bending was quite painless (1/10 NPRS); however, flexion and extension were painful (7/10 and 5/10 NPRS, respectively) and the ROM was limited (60° and 10°, respectively). The passive ROM was also assessed: extension with overpressure and forward flexion with overpressure exacerbated the patient’s back pain (8/10 and 7/10 NPRS, respectively). Both passive lateral side bending and passive trunk rotations resulted in pain worsening (3/10 NPRS and 4/10 NPRS, respectively). No strength abnormalities were reported in all planes.

The PT looked for some trigger points in the patient’s back muscles: these trigger points were present in trunk extensor, quadratus lumborum and gluteus; however, none of these reported the patient’s familiar pain.

The patient was assessed in prone position for accessory vertebral movements. Central and unilateral passive accessory intervertebral motion tests on L1 to S1 segment elicited the patient’s familiar pain, which became deep and widespread in the sacral area when the PT applied pressure on L4, L5 and S1. For a better assessment of the vertebral status, the PT performed the closed-fist percussion sign, which was positive as the patient complained of a sharp, sudden pain when PT hit the L4, L5 and S1 segment. In addition, the Supine Sign was administered and resulted positive, since the patient was unable to lie supine due to his LBP [21]. The PT administered the Italian versions of SF-36 [22] and the Oswestry Disability Index [23] to, respectively assess the patient’s quality of life and extent of disability (Table 1, baseline column).

## 4. Referral

Both anamnestic collection and physical examination revealed red flags [24], which suggested an extra-professional pathology. The PT wrote a letter describing the main features of the patient’s medical history and clinical examination and referred the patient for neurosurgical consultation.

The neurosurgeon examined the patient and prescribed an MRI, which revealed: “lumbar and sacral multiple HAs characterized by T2 hyperintensity, revealing major vascular component. Numerous HAs limited into the vertebral bone in L1, L4, L5 and S3 and not occupying the pedicles area. Wide HAs extended into the left vertebral pedicles in L2 and L3, without overcoming the border of the vertebral cortex. Presence of large HAs which occupy the entire body of S2, exceed the posterior cortex and occupy the medullary space” (Figure 3A,B).

The neurosurgeon, after the MRI evaluation, classified the HA as “Benign Vascular Tumour”, according to the ISSVA classification [1] and suggested an EVE, which was performed twenty days later.

## 5. Physiotherapy Intervention

One month after the EVE, the patient returned to the authors’ private practice for post-surgery rehabilitation, according to the neurosurgeon prescription.

During his rehabilitation behavioural education for daily living activities, manual therapy, therapeutic exercises, electrotherapy, strengthening and stretching modalities were performed. Table 2 showed the patient’s rehabilitation path.

## 6. Follow-Up

The patient’s range of motion of the trunk was measured two months after surgery. Flexion (90°), right and left lateral bending (40°) improved and were painless, while extension improved too (30°), but showed a 2/10 NPRS at end of range.

The Italian version of SF-36 and the Oswestry Disability Index were re-administered and their results are displayed in Table 1 (follow-up column).

## 7. Discussion 

This clinical case describes the history, evaluation and treatment of a patient with specific LBP due to aggressive vertebral HAs.

In accordance with clinical guidelines [25], LBP is usually considered a musculoskeletal disorder with a positive prognosis, commonly treated by PTs using education, manual therapy and exercises. However, in a low percentage of cases, LBP may result from an extra-physiotherapeutic pathology, such as neoplasm, infections, cauda equina syndrome or fractures [14,26].

The ability to perform an adequate screening of the patient and to recognize such pathologies is a key component of the physiotherapist’s practice [27,28], especially in the path of direct access to treatment. Differential diagnosis in PT practice is the result of a complex process of clinical reasoning and decision making, which integrates information from the patient’s history, physical examination and imaging findings, when available [29]. A thorough analysis of these information improves the likelihood of ruling in or ruling out the presence of red flags [30]. Notably, the patient’s medical history is the cornerstone in PT assessment [24]. Obtaining information about the clinical condition of a patient with apparent musculoskeletal disorders [31] facilitates the PT’s job and reduces the incidence of serious and misdiagnosed pathologies [32]. 

Since musculoskeletal symptoms can be the result of orthopaedic, neurological, or rheumatological processes, a comprehensive neuromuscular examination should always include a detailed neurological and musculoskeletal evaluation [33]. 

Furthermore, test outcomes are best interpreted in the context with the entire examination profile, where the sensitivity and specificity of these tests can affect their usefulness in detecting red flags [30,34].

In this case, during the anamnestic analysis, various pieces of information emerged which alarmed the PT: unjustified weight loss in the past weeks, pain always present and not improved with drugs, removal of a malignant tumour five years prior, smoking and high BMI. Moreover, the physical examination also revealed significant data: central and unilateral passive accessory intervertebral motion tests, positive closed-fist percussion sign and supine sign, which elicited and worsened the patient’s familiar pain.

RFs identification should be the starting point in the screening process development for patients with thoraco-lumbar pain; indeed, although the use of a single red flag is not recommended in clinical practice, the combination of multiple red flags presents a greater diagnostic accuracy [35].

For example, this case revealed findings that were not within the scope of the PT’s knowledge, experience or expertise, and referring the patient to another health professional was to be considered mandatory. PTs are among the most frequently sought-after healthcare professionals for pain and disability management in relation to musculoskeletal disorders [18] and have the responsibility to recognize when a patient referral/consultation is necessary [36]. Furthermore, direct access is recommended by the most up-to-date guidelines as a first-line treatment, due to its cost-effectiveness, safety and patient satisfaction compared to other interventions [37,38]—ensuring the possibility for the patient to directly seek a physiotherapist as first contact [11]. The evidence emerging from recent studies highlights the potential role of direct access to physiotherapy in reducing the costs associated with the care path: fewer visits and examinations and a more active approach, which can allow patients with musculoskeletal disorders to achieve an earlier and more functional recovery [11]. However, despite the PT’s ability to work independently from other health professionals, this case underlines how the collaboration between different professionals could be beneficial and such cooperation could lead to the best outcome for the patient.

## 8. Conclusions

In conclusion, this case emphasizes the importance of using in-depth qualitative interviews in order to gain a deeper vision on the presence of red flags in back pain assessment. Although their level of involvement in the pathological diagnosis has been debated, physical therapists must be able to screen for conditions outside the scope of their practice in order to refer their patients to the most appropriate professional; consequently, screening for referral is an essential skill and responsibility of all physical therapists.

Moreover, in this case report, a proper and fast clinical history assessment and physical examination led the patient to a prompt referral and a timely surgery, before his pathology worsened: the collaboration between health professionals was essential to allow the patient to continue his life safely.

## 9. Patient’s Point of View

The patient was satisfied with the work carried out by the interdisciplinary team, happy to have undergone EVE and to have followed the rehabilitation procedure described, which has changed his life for the better—both from a work and social point of view. The limitation of this evaluation was that no standardized questionnaire was administered to the patient to assess his degree of satisfaction.

## Figures and Tables

**Figure 1 ijerph-19-13276-f001:**
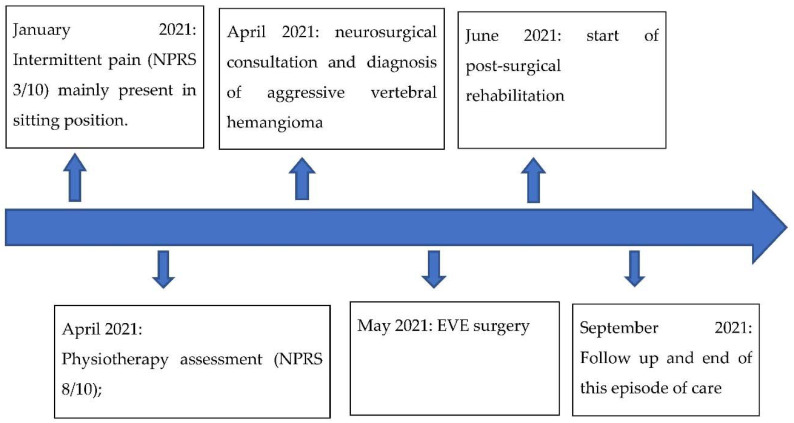
Timeline. Acronyms: *NPRS*, Numeric Pain Rating Scale; *EVE*, Endovascular Embolization.

**Figure 2 ijerph-19-13276-f002:**
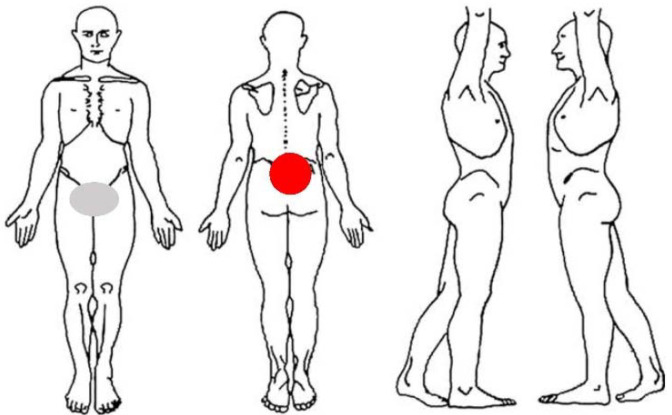
Body chart. Red circle indicates pain; grey circle indicates dysesthesia.

**Figure 3 ijerph-19-13276-f003:**
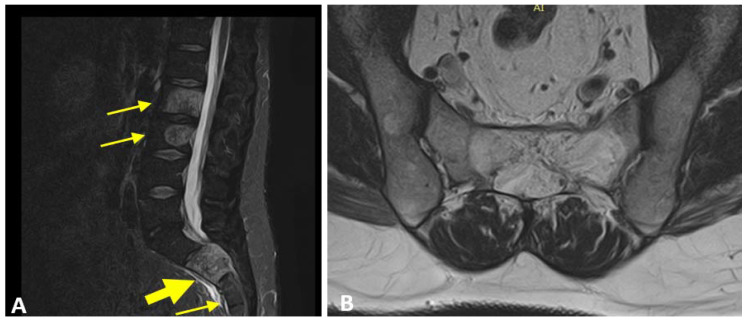
Multiple HAs were displayed in the lumbar and sacral vertebrae (little yellow arrows). S2 vertebral body (big yellow arrow) showed a wide HAs overcoming the cortex and occupying the medullary space ((**A**), sagittal view; (**B**), transversal view).

**Table 1 ijerph-19-13276-t001:** Clinical evaluation of patient. PA; Physical Activity; RP; Role Physical; BP; Bodily Pain; GH, General Health; VT, Vitality; SF, Social Functioning; RE, Role Emotional; MH, Mental Health.

SF-36 for health-related quality of life *	Baseline	Follow-Up
	PA = 50	PA = 95
	RP = 25	RP = 100
	BP = 22	BP = 84
	GH = 25	GH = 76
	VT = 10	VT = 70
	SF = 25	SF = 87
	RE = 0	RE = 100
	MH = 28	MH = 80
Oswestry Disability Index **	Baseline	Follow-up
	45	10

* Range 0–100 (0 = worst quality of life, 100 = best quality of life). ** Range 0–100 (0 = lower level of disability, 100 = higher level of disability).

**Table 2 ijerph-19-13276-t002:** Rehabilitation path.

Weeks 1–2	Patient education; Electrotherapy (paravertebral muscles stimulation); Basic core and paravertebral muscles isometric exercises; Lumbosacral soft mobilization; Soft proprioceptive–kinaesthetic exercises.
Weeks 3–4	Patient education; Myofascial treatment of lumbosacral trigger points; Basic core and paravertebral muscles strengthening exercises during electrotherapy (e.g., crunch, bird dog, plank, side plank); Lumbosacral mobilization; Proprioceptive–kinaesthetic exercises with different tools (unstable platforms, airbeds, …)
Weeks 5–6	Intermediate strengthening exercises for core and paravertebral muscles (e.g., boat, superman from the ground, bridge, etc.); Lumbo-sacral myofascial treatment of trigger points; Lumbosacral mobilization; Proprioceptive–kinaesthetic exercises.
Weeks 7–8	Myofascial treatment of trigger points; Advanced strengthening exercises (e.g., dead lift, barbell squat, etc.) also with different tools (elastic bands, dumbbells, cattle balls...)

## Data Availability

Data generated or analysed during this study are included in this published study. Other information of this study is available from the corresponding author on reasonable request.

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
