# Peer review of "Aggressive Vertebral Hemangioma and Spinal Cord Compression: A Particular Direct Access Case of Low Back Pain to Be Managed—A Case Report"

_ijerph, 2022, doi:10.3390/ijerph192013276_

Round 1

Reviewer 1 Report

The authors present here a case report highlighting the crucial role of Physiotherapists in assessing uncommon low back pain causes, such as vertebral hemangiomas.

The topic is interesting, nevertheless, I've some suggestions:

1) An extensive editing of the English language and style should be done, I suggest submitting the manuscript to a scientific and medical editing service.

2) The ISSVA classification of Vascular Anomalies should be used and cited.

3) Please give references for the "Numeric Pain Rating Scale", the "SF-36" and the "Oswestry Disability Index" questionnaires.

4) Nowadays, a profitable interdisciplinary collaboration among different healthcare professionals can be considered just good clinical practice. Please emphasize what's new in this case and how this case can help to manage similar cases in the future.

Reviewer 2 Report

This study, a case report, entitled "Aggressive vertebral hemangioma and spinal cord compression: a particular first access of low back pain to be managed" aimed to highlight the importance of screening for red flags and to report the physiotherapist’s clinical reasoning that led to refer patients to other healthcare professionals.

This is an important case report, however, some relevant concerns should be addressed by the authors:

- Improve the rationally. Why it this case important?

- In the case description, why was this participant selected? Is this a realu rare case?

- None mention about ethics procedures in the manuscript. Was this project submited to the Ethics Committee?

- In the page 7 the authors described patient's view. However, none mention in the text about questionnaires or interview. How was this information collected? How reability and valited is this information? How sure you are about the information that the "patient was satisfied with the work carried"?

Round 2

Reviewer 1 Report

I'm satisfied with the improvements made by the authors.